# Identifying Early Metabolic Risks of Obesity in Mexican Children and Adolescents from a Semi-Rural Community in Mexico: Beyond BMI and into Biochemical and Nutritional Markers

**DOI:** 10.3390/nu17132195

**Published:** 2025-06-30

**Authors:** Nurit Bistre, Sara Guillén-López, Isabel Medina-Vera, Miriam E. Reyna-Fabián, Nancy L. Hernández-Martínez, Lilian Castro-Monroy, René Cerritos-Flores, Ana Karen Arias-Basilio, Diana González-Santiesteban, Cynthia Fernández-Lainez, Marcela Vela-Amieva, Liliana Fernández-Hernández

**Affiliations:** 1Laboratorio de Biología Molecular, Instituto Nacional de Pediatría, Secretaría de Salud, Mexico City C.P. 04530, Mexico; nuritbistre@gmail.com (N.B.);; 2Laboratorio de Errores Innatos del Metabolismo y Tamiz, Instituto Nacional de Pediatría, Secretaría de Salud, Mexico City C.P. 04530, Mexico; guillen.lopez.sara@gmail.com (S.G.-L.);; 3Departamento de Metodología de la Investigación, Instituto Nacional de Pediatría, Secretaría de Salud, Mexico City C.P. 04530, Mexico; 4Independent Researcher, Mexico City C.P. 03630, Mexico; 5Comisión Intersecretarial de Bioseguridad de los Organismos Genéticamente Modificados, Secretaría de Ciencia, Humanidades, Tecnología e Innovación, Morelia C.P. 60170, Mexico

**Keywords:** pediatric obesity, micronutrient intake, dietary adequacy, early metabolic risk, anthropometric

## Abstract

**Background:** Childhood and adolescent obesity often coexist with micronutrient deficiencies and metabolic alterations, particularly in marginalized communities. **Objectives:** This cross-sectional study evaluated the biochemical, anthropometric, and dietary characteristics of 55 children and adolescents (ages 4–13) from Tlaltizapán, Mexico, to identify the early metabolic risk factors associated with excess weight. **Methods:** Nutritional intake was assessed through six-day dietary recalls and analyzed for adequacy against the national reference values. Anthropometric and biochemical indicators—including the BMI-for-age Z-score, waist-to-height ratio (WHtR), lipid profile, and plasma amino acid levels—were stratified by age and weight status. **Results:** Overall, 36.4% of participants were overweight or obese. Alarmingly, 89.4% of children and 94.1% of adolescents had low HDL levels, regardless of their BMIs. Several participants with a normal BMI showed elevated WHtR, triceps skinfold, and plasma branched-chain amino acids, suggesting hidden metabolic risks. The dietary analysis revealed an excess intake of sugars, proteins, and fats, alongside insufficient fiber, vitamins, and minerals. The adolescents had poorer adherence to dietary recommendations than children. **Conclusions:** These findings underscore the limitations of BMI alone and support the use of WHtR, skinfolds, and biochemical markers to detect preclinical obesity. Urgent, targeted nutritional strategies are needed in semi-rural areas to address the double burden of obesity and undernutrition.

## 1. Introduction

In Mexico, overweight and obesity remain highly prevalent across all age groups and regions, affecting both urban and rural populations. According to the 2020–2023 National Health and Nutrition Survey (ENSANUT), the prevalence of overweight and obesity among school-aged children was 36.5%, and 40.4% among adolescents [1].

Certain populations face compounded challenges related to poverty, malnutrition, and limited access to affordable, nutritious food, often coupled with sedentary lifestyles. This is especially critical in communities with high levels of marginalization. Although these issues are well-documented, studies specifically analyzing the nutritional status and obesity in marginalized Mexican populations remain scarce. The ENSANUT-100K study, for instance, reported that urban residence increases the risk of overweight by 22.4%, compared to 21.4% in rural areas [2].

In the state of Morelos, Mexico, rural and semi-rural communities experience high levels of food insecurity. Data from the 2018 ENSANUT revealed that 67.8% of Morelos residents face food insecurity, with even higher rates in rural areas (81.4%). This includes 43.2% of households with mild food insecurity, 23.1% with moderate, and 15% with severe levels [3].

Tlaltizapán de Zapata, a semi-rural municipality in Morelos, is primarily sustained by agricultural activity and is characterized by widespread poverty. According to the 2021 Municipal Statistical Summary, 59.1% of the local population lives below the poverty line [4]. Previous anthropometric data from Tlaltizapán reported that 20% of children were overweight, 15.6% had obesity, and 9.8% were undernourished, revealing a complex and worrisome nutritional profile [5]. Children and adolescents under 20 years of age exhibit high consumption of low-nutrient-dense food. Those aged 5 to 11 years show the greatest intake of snacks, sweets, sugary cereals, desserts, and sweetened dairy beverages [3]. This situation must be addressed more comprehensively by including biochemical and dietary parameters to better understand this complex phenomenon in rural communities.

Childhood and adolescence represent critical windows for establishing healthy eating behaviors and effective programmatic interventions [6]. The integral approach of studying anthropometric, biochemical, and dietetic indicators is important due to obesity significantly increasing the risk of type 2 diabetes, cardiovascular disease, hypertension, gastrointestinal and respiratory disorders, and psychosocial complications. It is also associated with metabolic alterations, such as dyslipidemia, hyperandrogenemia, hyperglycemia, and insulin resistance, all of which can progress to metabolic syndrome [7]. Furthermore, obesity has been linked to various micronutrient deficiencies and altered plasma amino acid profiles, which may result from poor dietary quality, nutrient malabsorption, or chronic low-grade inflammation [8,9]. Elevated levels of branched-chain amino acids (BCAAs) are positively associated with a person’s BMI and have been proposed as potential early indicators of metabolic disturbances [10]. Pediatric obesity is associated with non-communicable diseases that can affect children, adolescents, and eventually adults, leading to increased public healthcare costs. Moreover, it may be linked to a lower future income, worsening the situation of poverty in rural and semi-rural areas. Studying these communities is essential to identifying at-risk populations and to providing appropriate, multidisciplinary healthcare interventions [11].

This cross-sectional study aims to comprehensively evaluate the biochemical, anthropometric, and dietary profiles of children and adolescents in Tlaltizapán, Mexico, in order to identify the early metabolic risk factors associated with overweight and obesity in this vulnerable population. These findings may help inform future efforts to detect and address similar risks in other comparable communities.

## 2. Materials and Methods

### 2.1. Study Design and Participants

This cross-sectional study was conducted between November 2023 and July 2024. This study included 55 participants from the semi-rural municipality of Tlaltizapán de Zapata, Morelos, Mexico. Children aged 4–8 years and adolescents aged 9–13 years [12] were eligible for inclusion. Participants were invited during routine pediatric visits at the Centro Pediátrico de Investigación Comunitaria (CePIC) in Tlaltizapán, where medical and nursing staff explained the study and extended the invitation. A clinical history was obtained from each participant, including information on their family history of chronic degenerative diseases and obesity.

Written informed consent was obtained from the parents or legal guardians of all participants. In addition, children over 8 years of age provided written informed assent in accordance with the ethical guidelines for research involving minors. They also signed a privacy notice to ensure the confidentiality of the data collected.

### 2.2. Nutritional Evaluation

Parents and caregivers received initial training to complete a three-day, 24 h dietary recall that was submitted during the next follow-up visit. Caregivers, children, and adolescents were told to complete the recall together. A single trained dietitian reviewed each dietary record. When information was missing or unclear, the dietitian used food models and standard household utensils (e.g., cups and spoons) to help estimate the portion sizes and cooking methods. Dietary data were collected at two time points: baseline and six months later, providing a representative six-day dietary intake profile per participant.

The same dietitian analyzed all reported foods using Metabolic Pro^®^ software [13]. Nutrient intake was averaged across the six days to calculate the adequacy percentage (AP) for each nutrient. The AP was calculated by dividing the amount of a nutrient consumed by the Recommended Dietary Allowance (RDA) and multiplying it by 100.

The participants were stratified into two age groups (4–8 years and 9–13 years), and nutrient intake was compared against the RDA values for the Mexican population, according to sex and age [12]. The AP was calculated to assess nutritional adequacy (see Appendix A).

### 2.3. Anthropometric Evaluation

Anthropometric measurements were obtained at baseline by a single standardized dietitian. All measurements were taken in the morning or early afternoon, with the participants wearing underwear and no shoes. Weight was measured using a Tanita^®^ pediatric digital scale (BF-689, Tanita Health Equipment HK LTD. Unit 301-303 Wing on Plaza 3/F 62 Mody Road Tsimshatsui East Kowloon, Hong Kong), and height was measured with a SECA 213 portable stadiometer (SECA Deutschland, Medical Measuring Systems and Scales. Hammer Steindamm 3-25 22089, Hamburg, Germany), following the World Health Organization (WHO)’s protocols [14].

The Body Mass Index (BMI) was calculated as the weight (kg) divided by the height squared (m^2^) using WHO Anthro Plus software (v.1.0.4), and classified according to the WHO’s 2007 BMI-for-age Z-score criteria: underweight (<−2 SD), normal weight (−2 to +1 SD), overweight (>+1 SD), and obesity (>+2 SD) [15]. Stunting was defined as a height-for-age Z-score < −2 SD. For children younger than 5 years, a BMI Z-score of −2 or less indicated being underweight, a normal weight was defined as −1.99 to 0.99, +1 to +2 indicated being at risk of being overweight, +2 to +3 indicated being overweight, and +3 or more indicated obesity according to the WHO’s cut-offs.

The mid-upper arm circumference (MUAC) and tricep skinfold thickness (TST) were measured in duplicate at the midpoint on the dominant arm using a Gülick fiberglass tape and a Lange skinfold caliper, respectively, according to the WHO’s guidelines [14]. Measurements were interpreted using Frisancho percentiles [16]. MUAC values < 5th percentile indicated a risk of undernutrition; the 5th–95th percentiles were considered normal; and values > 95th percentile indicated obesity risk or muscle hypertrophy. The TST percentiles were interpreted as follows: the ≤5th percentile indicated muscle mass depletion, the 5th–15th indicated a low fat mass (at risk), the 15th–75th indicated an average fat mass, the 75th–85th indicated a high fat mass (at risk), and the >85th indicated excess fat mass or possible obesity.

The waist circumference (WC) was measured in duplicate according to the WHO’s protocols [17]. Based on the International Diabetes Federation [18], WC values above the 90th percentile were considered at risk. The waist-to-height ratio (WHtR) was also calculated, with values >0.5 indicating metabolic risk [19].

### 2.4. Biochemicals

Peripheral blood samples were collected after an 8 h fast. The tests included a complete blood count, fasting glucose, insulin, total cholesterol, LDL, HDL, triglycerides, liver transaminases (AST and ALT), and plasma amino acids. To this end, an analysis and processing of blood samples were carried out in a surrogate clinical laboratory under protocols that accomplished internationally recognized standards for quality. The reference ranges are presented in Appendix A. Abnormal values were defined as those falling outside the reference intervals. Anemia was defined as hemoglobin concentrations below 13.0 g/dL, based on the pediatric reference ranges established by the clinical laboratory. The biochemical parameters were categorized according to the participants’ BMIs (underweight, normal weight, overweight, and obesity) to assess statistical significance across these groups.

#### Plasma Amino Acids

Blood amino acids, including alanine, arginine, citrulline, glycine, XLE-OHPro (a composite of leucine, isoleucine, alloisoleucine, and hydroxyproline), methionine, ornithine, phenylalanine, proline, tyrosine, and valine, were quantified using tandem mass spectrometry following a standard protocol, as described by Ibarra-González [20]. A commercial kit (Neobase, Perkin Elmer, Wallac Oy, Turku, Finland) was used to extract amino acids from the blood samples, following the manufacturer’s instructions. The samples were then analyzed using a triple quadrupole tandem mass spectrometer (Quattro micro-API, Waters Inc., Milford, MA, USA) with electrospray ionization to separate, detect, and quantify the amino acids.

### 2.5. Statistical Analysis

Continuous variables are presented as the mean ± standard deviation or as the median (25th–95th percentile), depending on the distribution. Dichotomous variables are expressed as frequencies. The Kolmogorov–Smirnov test was used to assess the normality of variable distributions. For comparisons, either the paired Student’s *t*-test or the Wilcoxon signed-rank test was applied, as appropriate. Statistical analyses were performed using SPSS software (version 25, IBM Corp., Armonk, NY, USA), and the figures were created using GraphPad Prism (version 9, GraphPad Software, San Diego, CA, USA).

### 2.6. Ethical Considerations

This study was conducted in accordance with the Declaration of Helsinki, and the protocol was approved by the Research, Ethics, and Biosafety Institutional Committees of the Instituto Nacional de Pediatría (approval number 2023/018, approved on 18 May 2023). All participating families received nutritional and medical follow-ups.

## 3. Results

### 3.1. Study Population

A total of 55 participants were enrolled: 38 (69%) were children and 17 (31%) were adolescents. The average weight and height were 26.4 ± 8.4 kg and 122.8 ± 8.6 cm in the children group, and 39.5 ± 12.6 kg and 142.2 ± 11.3 cm in the adolescent group, respectively. Based on the BMI-for-age Z-scores of the 55 participants, 60% (33 children and adolescents) were of normal weight, 36.4% (20 children and adolescents) were classified as overweight or obese, and 3.6% (two adolescents) were underweight. Table 1 details the demographic characteristics of the population.

### 3.2. Biochemical Parameters

The distribution of lipid profile alterations according to weight status in both children and adolescents is summarized in Table 1. This table shows the number of participants with elevated total cholesterol, LDL, high triglycerides, or low HDL cholesterol, stratified by nutritional status (undernourished, normal weight, overweight, and obesity). Additional biochemical characteristics are provided in Table 2.

### 3.3. Anthropometric Indicators

Waist circumference above the 90th percentile was observed in 21.1% of the children and 23.5% of the adolescents. The anthropometric characteristics of the population are detailed in Table 3.

When the waist-to-height ratio (WHtR) was compared to the participants’ BMIs categories, it was found that in the children’s group, 7.8% of the participants with a normal BMI already had a WHtR > 0.5. Among those classified as obese, 31.5% had a WHtR > 0.5. In the adolescent group, 11.7% of the participants with overweight and 23.5% of those with obesity also showed a WHtR above 0.5 (Figure 1).

Regarding the tricep skinfold thickness (TST), 44.7% of the children had values above the 85th percentile, indicating excess fat mass. Notably, among these children, 18.4% had a normal BMI, 5.3% were overweight, and 21% were obese. In the adolescent group, 41.1% presented with a TST above the 85th percentile; of these, 17.6% were overweight and 23.5% were obese (Figure 2).

### 3.4. Family Medical History

Among participants, 61.8% had a family history of type 2 diabetes, 43.6% had a history of obesity, and 16.3% had a history of cardiovascular disease.

### 3.5. Differences in Adequacy Percentage of Nutrients Between Children and Adolescents

Our analysis revealed notable differences in the adequacy percentage of several nutrients between school-aged children and adolescents. The children met 78% of their recommended daily caloric intake, whereas the adolescents met only 55% of their age-specific RDA.

The adequacy protein percentage was significantly higher in children, who reported an ingestion of 226.9% of their RDA for age, compared to 148.2% in adolescents (*p* < 0.0001). The percentage of energy from protein was 16.5% and 17.4% in children and adolescents, respectively, above the recommended 15%. Amino acid intake was also significantly higher in the 4–8-year-old group (*p* < 0.05). These differences are summarized in Appendix A.

Fat intake was similar between groups, averaging 47.2 ± 10.9 g in children and 44.2 ± 10.4 g in adolescents. In both groups, the percentage of energy derived from fat exceeded the recommended limit; children obtained 33.9% and adolescents 34% of their total energy intake from fat, surpassing the 30% recommended by the Mexican Dietary Reference Intakes. Average cholesterol intake also exceeded the RDA of 130 mg per 1000 kcal, reaching 107% in children and 139.9% in adolescents. Intake of Omega-3 and Omega-6 fatty acids was below the recommended levels in both groups.

Carbohydrate intake averaged 156.8 ± 35.2 g in children and 138.9 ± 37.8 g in adolescents, representing 49.6 ± 4.8% and 48.1 ± 5.2% of the total energy intake, respectively. Fiber intake was suboptimal in both groups: children met only 56% of their RDA, while adolescents reached just 35.3%. In contrast, both groups consumed significantly more added sugars than recommended, with children exceeding the recommended limit by 147% (total intake: 247%) and adolescents by 140% (total intake: 240%).

Significant differences (*p* < 0.05) were observed in the intake adequacy of vitamins C, E, A, B6, and B12, with children consistently exhibiting higher adequacy percentages than the adolescents. Overall, the AP of most micronutrients and vitamins in both groups remained below the RDA. Key micronutrients, such as calcium, iron, and zinc, did not meet the RDA in either group. In contrast, phosphorus, selenium, and sodium reached or exceeded 100% of the RDA in children, while among adolescents, only selenium and sodium approached or surpassed the recommended values.

Anemia was detected in approximately one-third of the study population in both age groups. Regarding vitamin intake, the children showed an adequate or near-adequate intake (≥100% RDA) for vitamin B6, B12, thiamine, riboflavin, and niacin, while vitamins C, D, E, K, A, and folate fell below the recommended levels. In adolescents, the adequacy percentages were low for vitamins C, B12, D, E, K, thiamine, A, and folate.

### 3.6. Plasma Amino Acid Levels

Comparisons between participants with a normal weight and those with excess weight showed that the former group had higher dietary adequacy percentages for essential amino acids based on the RDA values: leucine (182.18%), isoleucine (179.8%), methionine (119.98%), phenylalanine (133.64%), valine (162.10%), and tyrosine (106.74%).

In contrast, participants with overweight or obesity had slightly lower adequacy percentages: leucine (177.04%), isoleucine (175.3%), methionine (118.97%), phenylalanine (126.39%), valine (155.61%), and tyrosine (105.26%).

Among children specifically, the plasma amino acid analysis revealed significantly higher concentrations of leucine/isoleucine (*p* = 0.010), valine (*p* = 0.001), phenylalanine (*p* = 0.016), tyrosine (*p* = 0.025), and proline (*p* = 0.003) in the overweight/obese group compared to those with a normal weight (Figure 3).

Additional differences in the plasma amino acid profiles between normal weight and overweight/obese children and adolescents are detailed in Appendix A.

## 4. Discussion

In this study, we evaluated the biochemical, anthropometric, and dietary profiles of children and adolescents, identifying the early metabolic risk factors associated with overweight and obesity, such as an altered waist-to-height ratio (WHtR) in both obese and normal weight individuals, low HDL cholesterol levels, elevated plasma amino acids, and notably, a high intake of protein, fat, cholesterol, sodium, and sugars, alongside a low intake of fiber, key vitamins, and minerals in a Mexican semi-rural cohort.

In this cohort, 20% of the participants reported hypercholesterolemia, 7.2% hypertriglyceridemia, and 5.4% hyperglycemia. A Mexican study of urban children aged 6–12 years reported higher rates of hypertriglyceridemia (22%) and below rates of hypercholesterolemia (12%) and hyperglycemia (4%), highlighting that the differences between rural and urban areas are becoming blurred [21]. Notably, 89.4% of children in our study exhibited low HDL cholesterol levels. This is consistent with existing data, indicating that the Mexican population, including children, frequently exhibits a dyslipidemic profile that is characterized by elevated triglycerides and reduced HDL levels. Previous studies have identified genetic contributors to these alterations. For example, polymorphisms in the *FTO*, *MC4R*, *LEP*, and *LEPR* genes have been significantly associated with obesity and hypertriglyceridemia in Mexican children aged 4–13 years (*n* = 718) [22]. Additionally, gene–gene and gene–diet interactions, such as those involving *SIDT2* and *ABCA1*, influence HDL cholesterol levels. The *ABCA1* rs9282541-A variant, for instance, was associated with lower HDL-c levels (OR = 1.34, *p* = 0.013) in Mexican adults (*n* = 1982) [23]. Further research is warranted to evaluate these genetic variants in the pediatric population of Tlaltizapán and their potential interaction with dietary patterns and ancestry. The observed high intake of dietary fat, protein, and unsaturated fatty acids—along with Mexican ancestry and the multifactorial nature of metabolic risk—may contribute to these metabolic disturbances, even in the absence of obesity.

The plasma amino acid analysis revealed that, although the concentrations of amino acids remained within the age-specific reference ranges provided by the laboratory, children with excess weight showed significantly higher levels of BCAAs—valine and leucine—as well as aromatic amino acids such as tyrosine and phenylalanine, and also proline, compared to those with a normal weight. These findings are consistent with previous literature linking altered amino acid profiles to obesity and insulin resistance [10,24]. The elevated BCAAs and aromatic amino acids may reflect a subclinical pro-inflammatory state, potentially driven by the consumption of foods rich in fat and protein, as suggested by the high intake levels observed in this group. Amino acid elevations have been implicated in the pathophysiology of obesity and are associated with increased metabolic risk, including the future development of insulin resistance [25,26]. In contrast, among adolescents, no statistically significant differences were found in the plasma amino acid concentrations between normal weight and overweight participants. This lack of significance is likely attributable to the smaller sample size in the adolescent group, which may have limited the statistical power.

Waist circumference and WHtR are increasingly recognized as key indicators of pediatric metabolic risk. WHtR, an index of central adiposity, is a useful predictor of cardiometabolic complications [27]. In our study, a WHtR > 0.5 was observed in children with a normal BMI, suggesting a possible hidden metabolic risk. One-third of children with obesity showed marked abdominal obesity. Among the adolescents, only those who were overweight or obese exhibited elevated WHtR values, which is indicative of increased cardiometabolic risk. Overall, 38% of the participants had a WHtR > 0.5, comparable to the 39% reported in an urban Mexican children study [21].

To gain a more comprehensive understanding of body composition, we compared the waist circumference and tricep skinfold thickness (TST) with the BMI. Our findings highlight the limitations of relying solely on the BMI to assess cardiometabolic risk. A notable number of children with a normal BMI displayed altered anthropometric and biochemical parameters, including an elevated TST, WHtR, and lipid abnormalities. These results highlight the importance of incorporating additional indicators to better evaluate adiposity and metabolic health in pediatric populations.

Based on the BMI-for-age Z-score interpretation, 36.4% of the participants were overweight or obese. These results are consistent with the 2022–2023 National Health and Nutrition Survey (ENSANUT), which reported a prevalence of 36.5% for overweight and obesity among school-aged children and 40.4% among adolescents in Mexico [1]. Similarly, Gatica-Domínguez (2019) reported that in Tlaltizapán, 35.6% of children were overweight or obese (20% overweight and 15.6% obese), a figure comparable to ours, though with a slight increase in total prevalence (36.4%), and a higher proportion of obesity (21.8%) compared to overweight (14.5%) [5].

In this context, the recent redefinition of obesity by The Lancet Diabetes & Endocrinology Commission, which differentiates preclinical and clinical obesity, becomes particularly relevant. Preclinical obesity refers to excess adiposity without current organ damage but with increased metabolic risk, while clinical obesity involves functional impairment [28]. Our results support this distinction, as several children with normal BMIs already exhibited early metabolic and anthropometric alterations. Early detection of preclinical obesity is crucial for implementing timely, preventive strategies, especially in semi-rural settings with limited access to specialized healthcare.

Our data also showed that participants with a familial history of obesity had lower adherence to RDA for age, which is consistent with prior evidence indicating that these individuals may face additional behavioral challenges [1,29].

Although we used 24 h dietary recalls instead of food frequency questionnaires—which limits our ability to assess the long-term dietary patterns—this method is culturally appropriate, accessible for low-literacy populations, and reliable for estimating the average group intake across multiple days [30]. One strength of this study was the thorough analysis of dietary data, which allowed a quantitative comparison with the RDA values. The low-energy intake observed in both children and adolescents may reflect underreporting and could potentially result in an overestimation of the micronutrient and vitamin deficiencies [31].

Alarmingly, 90.9% of our participants exceeded the WHO’s recommended limit of ≤5% of the total energy intake from added sugars [32]. The macronutrient distribution and AP for micronutrients and vitamins suggest a dietary pattern rich in animal products, saturated fats, and simple sugars, and low in fruits, vegetables, dairy, and wholegrains.

We also observed that adolescents had poorer adherence to RDA than younger children, which may be influenced by increasing autonomy and media exposure. A study from Andalusia [33] reported similar trends, showing lower intakes of calcium, magnesium, zinc, and vitamins B12 and E among adolescents. Likewise, a Brazilian study showed that a healthy dietary pattern was more common among children, while adolescents exhibited a more restrictive and sugar-heavy pattern [34].

Our findings reaffirm that overweight and obesity in children often coexist with macro- and micronutrient deficiencies, as previously described [9]. These deficiencies may be due not only to a low intake of nutrient-dense foods, as seen in our study, but also to impaired absorption and chronic low-grade inflammation [8]. Studies have shown that individuals with obesity who consume hypercaloric diets often have a poorer micronutrient status than those consuming isocaloric but higher quality diets [35].

## 5. Conclusions

Several nutritional metabolic risks were identified in children and adolescents, regardless of their BMI Z-score, including the anthropometric (WtHR), biochemical (HDL), and dietary (high fat, protein, less fiber, vitamins, and minerals) indicators. These findings emphasize the urgent need to move beyond conventional weight-based metrics. The higher obesity prevalence observed in this population, compared to previous reports from the same area, highlights the importance of early detection and intervention.

Together, these results support the need for a more nuanced and multifactorial approach—integrating anthropometric, biochemical, and dietary indicators—for the early identification of at-risk individuals, even among those with a normal BMI. This is particularly important for ensuring early and equitable access to prevention and personalized dietary interventions particularly in marginalized communities.

## Figures and Tables

**Figure 1 nutrients-17-02195-f001:**
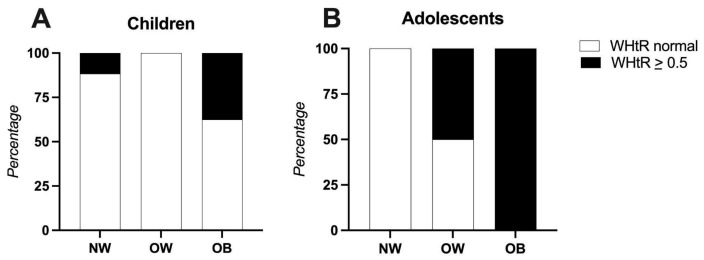
Waist-to-height ratio compared to participants’ BMIs. Panel (**A**): Children (4–8 years), Panel (**B**): Adolescents (9–13 years). Abbreviations: NW: normal weight, OB: obese, OW: overweight, WHtR: waist-to-height ratio.

**Figure 2 nutrients-17-02195-f002:**
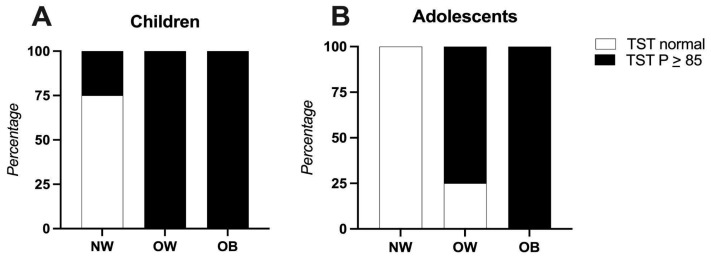
Tricep skinfold thickness compared to participants’ BMIs. Panel (**A**) Children (4–8 years), Panel (**B**): Adolescents (9–13 years). Abbreviations: NW: normal weight, OB: obese, OW: overweight, TST: tricep skinfold thickness.

**Figure 3 nutrients-17-02195-f003:**
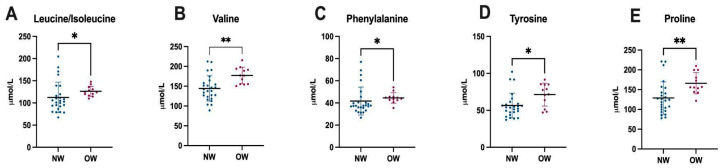
Differences in plasma amino acid levels between normal weight and overweight children. Abbreviations: NM: normal weight, OW: overweight, *: *p* ≤ 0.05, **: *p* ≤ 0.001.

**Table 1 nutrients-17-02195-t001:** Nutritional status distribution among children and adolescents with low HDL cholesterol and elevated levels of total cholesterol, LDL, and triglycerides.

	Children	Adolescents
	*n* = 38	*n* = 17
Elevated Total Cholesterol, *n* (%)		
Undernourished	0	0
Normal weight	2 (5.3%)	1 (5.9%)
Overweight	2 (5.3%)	1 (5.9%)
Obesity	3 (7.9%)	2 (11.8%)
Elevated LDL Cholesterol, *n* (%)		
Undernourished	0	0
Normal weight	1 (2.6%)	1 (5.9%)
Overweight	2 (5.2%)	1 (5.9%)
Obesity	1 (2.6%)	1 (5.9%)
Low HDL Cholesterol, *n*(%)		
Undernourished	0	2 (11.7%)
Normal weight	24 (63.1%)	6 (35.4%)
Overweight	3 (7.9%)	4 (23.5%)
Obesity	7 (18.4%)	4 (23.5%)
High Triglycerides, *n* (%)		
Undernourished	0	0
Normal weight	0	1 (5.9%)
Overweight	0	0
Obesity	2 (5.3%)	1 (5.9%)

Abbreviations: HDL: high-density lipoprotein, LDL: low-density lipoprotein.

**Table 2 nutrients-17-02195-t002:** Demographic, anthropometric, and biochemical characteristics of children and adolescents from Tlaltizapán, Mexico.

Descriptive Characteristics of the Population		
	**Children** ***n* = 38**	**Adolescents** ***n* = 17**
Age ± SD	6.9 ± 0.9	10.1 ± 1.2
Gender, *n* (%) (Female) (Male)	20 (52.6) 18 (47.4)	10 (58.8) 7 (41.2)
Z-score BMI interpretation, *n* (%) (Underweight) (Normal) (Overweight) (Obese)	0 26 (68.4) 4 (10.5) 8 (21.1)	2 (11.8) 7 (41.2) 4 (23.5) 4 (23.5)
**Biochemical and Metabolic Alterations**		
Anemia, *n* (%)	12 (31.6)	4 (26.5)
High Glucose levels, *n* (%)	2 (5.3)	1 (5.9)
Elevated Total Cholesterol, *n* (%)	7 (18.4)	4 (23.5)
Elevated LDL Cholesterol, *n* (%)	4 (10.5)	3 (17.6)
HDL, *n* (%) (Low) (Borderline low) (Optimal)	12 (31.6) 22 (57.9) 4 (10.5)	5 (29.4) 11 (64.7) 1 (5.9)
Elevated Triglycerides, *n* (%)	2 (5.3)	2 (11.8)
Elevated AST, *n* (%)	11 (28.9)	2 (11.8)
Elevated ALT, *n* (%)	2 (5.3)	0 (0)

Abbreviations: AST: Aspartate Aminotransferase, ALT: Alanine Aminotransferase, BMI: Body Mass Index, HDL: high-density lipoprotein, LDL: low-density lipoprotein, SD: standard deviation.

**Table 3 nutrients-17-02195-t003:** Nutritional interpretation of height-for-age, mid-arm circumference, tricep skinfold thickness, waist circumference, and waist-to-height ratio with categorical distributions.

Z Score Height for age interpretation, *n* (%)		
(Normal)	36 (97.3)	15 (88.2)
(Stunted)	1 (2.6)	2 (11.8)
Mid-arm-circumference interpretation, *n* (%)		
(Normal)	25 (65.8)	13 (76.5)
(Underweight risk)	10 (26.3)	3 (17.6)
(Obesity risk or muscle hypertrophy)	3 (7.9)	1 (5.9)
Triceps skinfold thickness interpretation, *n* (%)		
(FM—depletion)	1 (2.6)	2 (11.8)
(FM below media—Risk)	1 (2.6)	1 (5.9)
(Average MM)	12 (31.6)	5 (29.4)
(FM above media—At risk)	7 (18.4)	2 (11.8)
(FM Excess—obesity)	17 (44.7)	7 (41.2)
Waist circumference, (cm)	61.1 ± 11	69 ± 13
Waist circumference interpretation, *n* (%)		
At risk	8 (21.1)	4 (23.5)
Waist-to-height ratio (media)	0.49 ± 0.07	0.48 ± 0.07
Waist-to-height ratio interpretation, *n* (%)		
At risk	15 (39.8)	6 (35.3)

Abbreviations: FM: fat mass.

## Data Availability

The original contributions presented in this study are included in the article. Further information can be available after inquiries to the corresponding author.

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
