# Peer review of "Identifying Early Metabolic Risks of Obesity in Mexican Children and Adolescents from a Semi-Rural Community in Mexico: Beyond BMI and into Biochemical and Nutritional Markers"

_nutrients, 2025, doi:10.3390/nu17132195_

Round 1

Reviewer 1 Report

Comments and Suggestions for Authors

The manuscript from Bistre-Tajfed titled “Identifying Early Metabolic Risks of Obesity in Children and Adolescents from Tlaltizapán, Mexico: Beyond BMI and into Biochemical and Nutritional Markers” aimed to evaluate the biochemical, anthropometric, and dietary characteristics of 55 children and adolescents (ages 4–13) from Tlaltizapán, Mexico, to identify early metabolic risk factors associated with excess weight.

General comment

In general, although with some important limitations, the study is well done, well described and also quite complete. However, I believe that the results can only marginally be used on a broad spectrum. These findings suggest that it is important to analyze in detail many biochemical, anthropometric and lifestyle parameters, but this means that in every area, in every region, for every population with its own characteristics this study or a similar one should be done.

Major

L101: Is it correct that after 8 years of age informed consent is not required? Does this mean that authorization from an ethics committee is not necessary?

Paragraph 2.3 Anthropometric evaluation: based on anthropometric data and reference parameters, the authors evaluate the percentage of risk of obesity, metabolic problems, etc. Are there differences related to sex?

L138: Why do the authors associate excess muscle mass with obesity?

Paragraph 3.2: I think it would be easier for the reader if the data were also presented in a table.

Figure 2 it is not visible and therefore not assessable

L289-290: In the case of both proteins and fats (as written below), the intake is not that far from the recommended one. I therefore do not understand the meaning of these results and their role, which is not clear even in the discussion.

L321-324: Are these differences significant?

L368-369: Why did the authors not consider these genetic aspects?

L384: was it statistically different than what was recommended?

L390: Did the authors try to estimate the statistical power of the sample size a posteriori?

Author Response

Reviewer 1:

The manuscript from Bistre-Tajfed titled “Identifying Early Metabolic Risks of Obesity in Children and Adolescents from Tlaltizapán, Mexico: Beyond BMI and into Biochemical and Nutritional Markers” aimed to evaluate the biochemical, anthropometric, and dietary characteristics of 55 children and adolescents (ages 4–13) from Tlaltizapán, Mexico, to identify early metabolic risk factors associated with excess weight.

General comment

In general, although with some important limitations, the study is well done, well described and also quite complete. However, I believe that the results can only marginally be used on a broad spectrum. These findings suggest that it is important to analyze in detail many biochemical, anthropometric and lifestyle parameters, but this means that in every area, in every region, for every population with its own characteristics this study or a similar one should be done.

Thank you for your thoughtful comments and for acknowledging the quality and completeness of the study. We fully agree with your observation regarding the contextual limitations of the findings and the need for localized assessments based on each population’s specific characteristics. In response to your suggestion, we have modified the manuscript title to better reflect the scope and generalizability of the study. The updated title now reads:

“Identifying Early Metabolic Risks of Obesity in Mexican Children and Adolescents from a Semi-Rural Community in Mexico: Beyond BMI and into Biochemical and Nutritional Markers.”

Major

Reviewer's comment 1: L101: Is it correct that after 8 years of age informed consent is not required? Does this mean that authorization from an ethics committee is not necessary?

Response: Thank you for your comment. We acknowledge that the wording in the original manuscript may have caused confusion. We would like to clarify that all parents or legal guardians of the participants signed an informed consent form, and children over 8 years of age additionally signed an informed assent form. Furthermore, the study was approved by the Institutional Ethics Committee under approval number 2018/023.

We changed the original text: “All participants and their parents or legal guardians provided written informed consent or assent (in the case of children older than 8 years)”.

For: “Written informed consent was obtained from the parents or legal guardians of all participants. In addition, children over 8 years of age provide written informed assent in accordance with ethical guidelines for research involving minors”, Lines 104-106.

Reviewer's comment 2: Paragraph 2.3 Anthropometric evaluation: based on anthropometric data and reference parameters, the authors evaluate the percentage of risk of obesity, metabolic problems, etc. Are there differences related to sex?

Response: Thank you for your comment. We did not find any statistically significant or clinically relevant differences between males and females that warranted inclusion in the results section.

Reviewer's comment 3: L138: Why do the authors associate excess muscle mass with obesity?

Response: Thank you for your observation. You are correct — mid-upper arm circumference may reflect either increased fat mass or muscle hypertrophy, while triceps skinfold thickness is more directly associated with fat mass. We have revised the text accordingly to clarify this distinction.

We have changed the original text:MUAC values <5th percentile indicated risk of  undernutrition; 5th–95th percentiles were considered normal; and values >95th percentile indicated obesity risk. TST percentiles were interpreted as follows: ≤5th percentile indicated muscle mass depletion, 5th–15th low muscle mass (at risk), 15th–75th average muscle mass, 75th–85th high muscle mass (at risk), and >85th excess muscle mass (obesity)”

For: “MUAC values <5th percentile indicated risk of undernutrition; 5th–95th percentiles were considered normal; and values >95th percentile indicated obesity risk or muscle hypertrophy. TST percentiles were interpreted as follows: ≤5th percentile indicated fat mass depletion, 5th–15th low fat mass (at risk), 15th–75th average fat mass, 75th–85th high fat mass (at risk), and >85th excess fat mass or possible obesity”, Lines 144-149 .

Reviewer's comment 4: Paragraph 3.2: I think it would be easier for the reader if the data were also presented in a table.

Response: Thank you for your valuable suggestion. We have created a summary table to present the data more clearly and facilitate reader comprehension. The corresponding paragraph has been revised to refer to this new table. Also we have changed the text.

We have changed the original text: “In children with elevated LDL levels, 5.2% were overweight and 2.6% were obese. Additionally, 5.3% of children with high triglycerides were obese. A striking 89.4% of children had low HDL cholesterol levels; 63.1% had normal weight, while 26.3% were overweight or obese. Among adolescents, 11.8% had high triglycerides, equally divided between those with normal weight and those with obesity. Remarkably, 94.1% had low HDL cholesterol levels, while only 5.9% had optimal HDL values. Among adolescents with low HDL, 11.7% were undernourished, 35.4% had normal weight, and 47% were overweight or obese”.

For: “The distribution of lipid profile alterations according to weight status in both children and adolescents is summarized in Table 1. This table shows the number of participants with elevated total cholesterol, LDL, high triglycerides, or low HDL cholesterol, stratified by nutritional status (undernourished, normal weight, overweight, and obesity)”, Lines 200-204.

Table 1. Nutritional status distribution among children and adolescents with low HDL cholesterol and elevated levels of total cholesterol, LDL, and triglycerides

Children

n= 38

Adolescents

n= 17

Elevated Total Cholesterol, n (%)

Undernourished

0

0

Normal weight

2 (5.3%)

1 (5.9%)

Overweight

2 (5.2%)

1 (5.9%)

Obesity

3 (7.9%)

2 (11.8%)

Elevated LDL Cholesterol, n  (%)

Undernourished

0

0

Normal weight

1 (2.6%)

1 (5.9%)

Overweight

2 (5.3%)

1 (5.9%)

Obesity

1 (2.6%)

1 (5.9%)

Low HDL Cholesterol, n  (%)

Undernourished

0

2 (11.7%)

Normal weight

24 (63.1%)

6 (35.4%)

Overweight

3 (7.9%)

4 (23.5%)

Obesity

7 (18.4%)

4 (23.5%)

High Triglycerides, n  (%)

Undernourished

0

0

Normal weight

0

1 (5.9%)

Overweight

0

0

Obesity

2 (5.3%)

1 (5.9%)

Abbreviations;  HDL: High-Density Lipoprotein, LDL: Low-Density Lipoprotein.

Reviewer's comment 5: Figure 2 it is not visible and therefore not accessible.

Response: Thank you very much for your observation, and we apologize for the omission. We have now correctly inserted Figure 2 in the revised version of the manuscript in the appropriate section (Line 260) for proper evaluation.

Reviewer's comment 6: L289-290: In the case of both proteins and fats (as written below), the intake is not that far from the recommended one. I therefore do not understand the meaning of these results and their role, which is not clear even in the discussion.

Response: Thank you for your valuable comment. While the percentage of total energy from protein and fat may appear close to the recommended range, we intended to highlight the relative increase in both of these macronutrients alongside a lower percentage of energy from carbohydrates. Furthermore, the protein intake notably exceeded the age-specific RDA, which we consider important as it reflects an imbalance in overall macronutrient distribution.

Reviewer's comment 7: L321-324: Are these differences significant?

Response: Thank you for your question. The differences presented in Supplementary Table 4 were not statistically significant; therefore, we decided to include them in the supplementary material rather than in the main text. However, we considered them relevant enough to report, as they may suggest trends worth exploring in future studies. It is important to note that our sample size may have limited the statistical power to detect significant differences. We believe that future studies with larger cohorts will be able to better evaluate these trends.

Reviewer's comment 8: L368-369: Why did the authors not consider these genetic aspects?

Response: Thank you for this observation. We acknowledge the significant role that genetic factors play in the development of metabolic disorders. Indeed, several studies in the literature have reported associations between specific genetic polymorphisms and traits such as overweight, hypertriglyceridemia, and low HDL levels in the Mexican population. While the present study focused on biochemical and nutritional aspects, we fully recognize the importance of integrating genetic data. Therefore, we plan to replicate these genetic studies in the future by analyzing risk-associated polymorphisms in the same population we studied.

Reviewer's comment 9: L384: was it statistically different than what was recommended?

Response: Thank you very much for your comment. We have clarified in the manuscript that while the plasma amino acid concentrations remained within the age-specific reference ranges provided by the laboratory, we observed statistically significant differences between children with overweight/obesity and those with normal weight. This highlights the relevance of subtle metabolic changes even within clinically normal ranges.

We have changed the original text: “Plasma amino acid analysis revealed significantly higher concentrations of BCAAs—valine and leucine—as well as aromatic amino acids such as tyrosine and phenylalanine, and also proline, in children with excess weight. These findings are consistent with previous literature linking altered amino acid profiles to obesity and insulin resistance”. 

For: “Plasma amino acid analysis revealed that, although the concentrations of amino acids remained within the age-specific reference ranges provided by the laboratory, children with excess weight showed significantly higher levels of BCAAs—valine and leucine—as well as aromatic amino acids such as tyrosine and phenylalanine, and also proline, compared to those with normal weight. These findings are consistent with previous literature linking altered amino acid profiles to obesity and insulin resistance”.

Reviewer's comment 10: L390: Did the authors try to estimate the statistical power of the sample size a posteriori?

Response: Thank you for your observation. Following your suggestion, we have now calculated the post hoc statistical power of our sample. The power for the children’s group was 96.5%, whereas for the adolescents it was 39.5%. We acknowledge that the smaller sample size in the adolescent group limited the statistical power, which we had already noted as a limitation of the study in the Discussion section.

Reviewer 2 Report

Comments and Suggestions for Authors

Dear Authors,

The article addresses a significant topic as the double burden of obesity and malnutrition remains a public health challenge particularly in low- and middle-income settings. The findings that several children with normal BMI show early metabolic and anthropometric alterations underscore the relevance of identifying preclinical obesity as a clinical stage which would inform targeted interventions especially in setting where access to specialized care is limited.

Please consider my comments bellow.

Lines 80-81: in reference 10, the authors do not explicitly state that elevated blood concentrations of branched-chain amino acids (BCAAs) are considered definitive early markers of metabolic risk. Rather, they suggest that increased BCAA levels may indicate early metabolic disturbances. Please consider adjusting the text accordingly.

Line 97: WHO defines adolescence as individuals ages 10-19 years. Could you please clarify according to which definition did you classify 9-13 years as adolescence?

Line 98: could you please clarify the inclusion criteria used for participant selection in you study? Was age the only criterion, or were there additional factors considered? Could you also please clarify how were the participants contacted or recruited for the study?

Lines 125-129: thank you for the detailed description. I would like to ask, however, how you handled data from 4 year old children, given that BMI z score cut off values for the 2-5 year age group differ from those for children aged 5-19 years.

Line 299: please include a description of how anemia was assessed in the Materials and Methods section.

Line 305: Please consider revising the phrase to “Comparisons between participants with normal weight and participants with excess weight” to align with person first language guidelines.

Author Response

Reviewer 2:

Dear Authors,

The article addresses a significant topic as the double burden of obesity and malnutrition remains a public health challenge particularly in low- and middle-income settings. The findings that several children with normal BMI show early metabolic and anthropometric alterations underscore the relevance of identifying preclinical obesity as a clinical stage which would inform targeted interventions especially in settings where access to specialized care is limited.

Please consider my comments below.

Reviewer's comment 1: Lines 80-81: in reference 10, the authors do not explicitly state that elevated blood concentrations of branched-chain amino acids (BCAAs) are considered definitive early markers of metabolic risk. Rather, they suggest that increased BCAA levels may indicate early metabolic disturbances. Please consider adjusting the text accordingly.

Response: Thank you for your observation. We agree that the original wording may have overstated the conclusion of reference 10. We have revised the sentence to better reflect the authors’ interpretation. The updated version now reads: “Elevated levels of branched-chain amino acids (BCAAs) are positively associated with BMI and have been proposed as potential early indicators of metabolic disturbances.”, Lines: 80-82.

Reviewer's comment 2: Line 97: WHO defines adolescence as individuals ages 10-19 years. Could you please clarify according to which definition you classify 9-13 years as adolescence?

Response: Thank you for your observation. You are correct that the WHO defines adolescence as the age range of 10 to 19 years. However, in our study, the age groups were determined based on the Mexican Dietary Reference Intakes (DRIs), which classify individuals into specific age categories, including the 9–13 years group. This classification allowed us to align our dietary analysis with national reference standards and ensure comparability with other local nutritional studies. We have added the corresponding reference in the text to clarify the basis for these age categories.

Reviewer's comment 3: Line 98: could you please clarify the inclusion criteria used for participant selection in your study? Was age the only criterion, or were there additional factors considered? Could you also please clarify how the participants were contacted or recruited for the study?

Response: Thank you for your question. We have now clarified the inclusion criteria and recruitment process in the Materials and Methods section. Participants were selected based on age and residency in the municipality of Tlaltizapán, Morelos. Children and adolescents receiving pediatric care at the Centro Pediátrico de Investigación Comunitaria (CePIC) in Tlaltizapán were invited to participate by the attending medical and nursing staff.

Inclusion criteria included: (1) children aged 4–13 years whose parents or legal guardians provided informed consent and who, in the case of children over 8 years old, also gave assent; and (2) residence in Tlaltizapán for at least the past two years, with no plans to move during the study period. This information has been added to the Methods section for clarity.

We have modified the original text: “The study included 55 participants from the semi-rural municipality of Tlaltizapán de Zapata, Morelos, Mexico. Children aged 4–8 years and adolescents aged 9–13 years were eligible for inclusion.”

For: “The study included 55 participants from the semi-rural municipality of Tlaltizapán de Zapata, Morelos, Mexico. Children aged 4–8 years  and adolescents aged 9-13 years were eligible for inclusion. Participants were invited during routine pediatric visits at the Centro Pediátrico de Investigación Comunitaria (CePIC) in Tlaltizapán, where medical and nursing staff explained the study and extended the invitation.”. Lines: 97-101.

Reviewer's comment 4: Lines 125-129: thank you for the detailed description. I would like to ask, however, how you handled data from 4 year old children, given that BMI z score cut off values for the 2-5 year age group differ from those for children aged 5-19 years.

Response: Yes, thank you for your accurate and specific observation, we had only just one 4 year-old participant that was reported with a BMI Z score of 0.09. Hence, we will include in anthropometric methodology the following sentence:

Lines 137-140: “For children younger than 5 years, a BMI Z-score of -2 or less indicated underweight, normal weight was defined as -1.99 to 0.99, +1 to +2 indicated at risk of overweight, +2 to +3 indicated overweight, and +3 or more indicated obesity according to WHO cut-offs”.

Reviewer's comment 5: Line 299: please include a description of how anemia was assessed in the Materials and Methods section.

Response: Thank you for pointing this out. We have now included a description in the Materials and Methods section clarifying that: “anemia was defined as hemoglobin concentrations below 13.0 g/dL, based on pediatric reference ranges established by the clinical laboratory performing the analyses”. Lines 161-163

The specific reference values are provided in Supplementary Table 2.

Reviewer's comment 6: Line 305: Please consider revising the phrase to “Comparisons between participants with normal weight and participants with excess weight” to align with person first language guidelines.

Response: Thank you for your suggestion. We agree with the importance of using person-first language and have revised the sentence accordingly. The updated version now reads: “Comparisons between participants with normal weight and those with excess weight showed that the former group had higher dietary adequacy percentages for essential amino acids based on RDA values.”, Lines: 327-329.

Round 2

Reviewer 1 Report

Comments and Suggestions for Authors

no comment